# Pancreatic Stone Protein as a Biomarker for Sepsis at the Emergency Department of a Large Tertiary Hospital

**DOI:** 10.3390/pathogens11050559

**Published:** 2022-05-09

**Authors:** Titus A. P. de Hond, Jan Jelrik Oosterheert, Susan J. M. van Hemert-Glaubitz, Ruben E. A. Musson, Karin A. H. Kaasjager

**Affiliations:** 1Department of Internal Medicine and Acute Medicine, University Medical Centre Utrecht, Utrecht University, 3584 CX Utrecht, The Netherlands; shemert@umcutrecht.nl (S.J.M.v.H.-G.); h.a.h.kaasjager@umcutrecht.nl (K.A.H.K.); 2Department of Internal Medicine and Infectious Diseases, University Medical Centre Utrecht, Utrecht University, 3584 CX Utrecht, The Netherlands; j.j.oosterheert@umcutrecht.nl; 3Central Diagnostic Laboratory, University Medical Centre Utrecht, Utrecht University, 3584 CX Utrecht, The Netherlands; r.e.a.musson-2@umcutrecht.nl

**Keywords:** pancreatic stone protein, reg1a, sepsis, biomarker, emergency department

## Abstract

Early recognition of sepsis is essential for improving outcomes and preventing complications such as organ failure, depression, and neurocognitive impairment. The emergency department (ED) plays a key role in the early identification of sepsis, but clinicians lack diagnostic tools. Potentially, biomarkers could be helpful in assisting clinicians in the ED, but no marker has yet been successfully implemented in daily practice with good clinical performance. Pancreatic stone protein (PSP) is a promising biomarker in the context of sepsis, but little is known about the diagnostic performance of PSP in the ED. We prospectively investigated the diagnostic value of PSP in such a population for patients suspected of infection. PSP was compared with currently used biomarkers, including white blood cell count (WBC) and C-reactive protein (CRP). Of the 156 patients included in this study, 74 (47.4%) were diagnosed with uncomplicated infection and 26 (16.7%) patients with sepsis, while 56 (35.9%) eventually had no infection. PSP was significantly higher for sepsis patients compared to patients with no sepsis. In multivariate regression, PSP was a significant predictor for sepsis, with an area under the curve (AUC) of 0.69. Positive and negative predictive values for this model were 100% and 84.4%, respectively. Altogether, these findings show that PSP, measured at the ED of a tertiary hospital, is associated with sepsis but lacks the diagnostic performance to be used as single marker.

## 1. Introduction

Sepsis is a clinical syndrome characterized by life-threatening organ dysfunction due to a dysregulated host response to an infection [1]. Despite efforts over the past decades to improve outcomes, sepsis still has high morbidity and mortality [2,3]. Mortality for sepsis is estimated to be between 20 and 40%, and long-term complications are frequent, including kidney failure, liver failure, depression, and neurocognitive impairment [4,5,6]. In addition, the worldwide incidence of sepsis is still increasing, leading to a high socioeconomic burden and pressure on healthcare systems [7].

To improve outcomes, early recognition of sepsis is essential [1]. For every hour of treatment delay, mortality is estimated to rise by approximately 8% [8]. On the other hand, it is suggested that aggressive therapy strategies lead to higher rates of unnecessary antibiotic use, resulting in increasing Clostridioides difficile infections. Additionally, complications of aggressive resuscitation include hyperchloremic metabolic acidosis and peripheral edema [9,10]. Therefore, early identification strategies should not only be fast, easy, and sensitive but also specific. The emergency department (ED) is usually the first echelon within the hospital where acutely ill patients are clinically assessed. Therefore, the ED plays a critical role in the early identification of sepsis. Nonetheless, diagnosing sepsis can be challenging due to the heterogeneity of clinical symptoms and limited time. Additionally, there is no gold standard test to diagnose sepsis, making the diagnosis subjective with high interobserver variability [11].

Biomarkers can assist clinicians in recognizing sepsis as early as possible. Over the past decades, more than 250 potential biomarkers have been identified, but only a few are useful in clinical practice [12]. Nowadays, only C-reactive protein (CRP) and leukocytes are widely implemented in healthcare systems [12,13]. Nonetheless, these markers are known to lack diagnostic accuracy, resulting in an ongoing need for reliable, fast, and easily accessible biomarkers [14,15]. Pancreatic stone protein (PSP), previously known as lithostathine or regenerating protein 1 alpha, might be a suitable biomarker for sepsis [16,17,18]. PSP is a 144-amino-acid glycoprotein, the exact function of which has not been elucidated yet [19]. PSP is associated with pancreatic inflammation, and levels are increased in patients with type II diabetes. More importantly, in the context of sepsis, there are several findings that indicate that PSP is involved in the defense mechanisms during the early phase of sepsis. For instance, PSP was elevated in patients with posttraumatic infections and infectious complications of burn wounds [18,20]. In these studies, PSP was shown to be related to the severity of inflammation and was able to activate neutrophil granulocytes by upregulating activation markers CD11b and CD62L [18]. Aside from neutrophil activation, PSP by itself is known to have antibacterial functions. PSP is able to induce bacterial aggregation which might be beneficial for preventing bacteria from penetrating the intestinal barrier in the gut [21]. Lastly, PSP levels were shown to start increasing up to 72 h before the clinical symptoms of sepsis appeared. Therefore, PSP might be helpful for diagnosing sepsis before patients start to become critically ill [20]. Altogether, its association with severity of inflammation, antimicrobial involvement, and its capability to rise during early infection make PSP an interesting biomarker for early sepsis recognition.

There have been several studies on the diagnostic value of PSP for sepsis, showing promising results [22,23,24]. However, all these studies were limited concerning their clinical applicability in an ED setting. For example, most studies were performed at an intensive care unit (ICU), and little is known about the performance of PSP in the ED. Additionally, most studies were performed in unselected cohorts, while, from a clinical perspective, it is of interest whether PSP can identify sepsis in patients that are suspected of an infection [25]. Additionally, a previous study showed a lack of performance of PSP concerning sepsis in neutropenic patients [26]. Given the increase in patients treated with immunosuppressive medication [27], it is, therefore, unclear what diagnostic performance PSP might have in the setting of a modern ED. To investigate this, we tested the diagnostic performance of PSP for patients suspected of an infection at the ED and compared it with that of C-reactive protein (CRP) and white blood cell count (WBC).

## 2. Materials and Methods

### 2.1. Study Design

This was a mono-center, semi-prospective, observational cohort study. The study was embedded in the SPACE cohort [28]. This is an ongoing patient cohort in the ED of the UMCU since September 2016 in which all consecutive internal medicine patients with a suspected infection in the ED are registered. The SPACE cohort was reviewed and approved by the Medical Ethical Committee of the UMCU under number 16/594 and registered in the Dutch Trial Register (NTR) under number 6916. Given the nature of this study, an exemption for written informed consent was obtained.

### 2.2. Study Population and Data Collection

All adult patients that presented at the internal medicine department (or subspecialties) with suspicion of infection from August to November 2021 were included. All patients were given standard laboratory blood tests. From this blood, residual samples were used for the PSP analysis. Baseline and follow-up data were extracted from the data registered in the SPACE cohort, the exact structure of which is described elsewhere [28]. Buderer’s formula for diagnostic research was used for sample size calculation. Based on a prevalence for sepsis of 20% in our cohort, we calculated an estimated sample size of 193 patients (with 20% dropout rate).

### 2.3. Laboratory Measurements

Standard laboratory blood tests included electrolytes, kidney function, liver enzymes, and hematological measurements. PSP values were measured with the CE-marked IVD PSP capsule on abioSCOPE^®^ platform (Abionic SA, Epalinges, Switzerland). This is a nanofluidic immunoassay technology which measures a point-of care PSP value within 8 min via a fingerstick test. The abioSCOPE platform can measure PSP values of up to 600 ng/mL [29]. Higher values were displayed as >600 ng/mL. PSP values were compared with routinely measured biomarkers CRP and WBC.

### 2.4. Patient Classification

Initially, the cohort was divided into three groups: no infection, uncomplicated infection, and sepsis. For every patient in the SPACE cohort, likelihood of infection was assessed using a predefined, four-point scale (ascending from none, possible, probable, to definite), as described by the Centers for Disease Control and International Sepsis Forum Consensus Conference [30,31]. Likelihood was based on clinical symptoms in combination with diagnostic tests such as cultures, laboratory results, and radiographic findings. Sepsis was defined as modified early warning score (MEWS) ≥ 5 (Appendix A) [32] in combination with a probable or definite infection. Uncomplicated infection was defined as a probable or definite infection with MEWS < 5. All other patients were defined as no infection due to limited or no evidence for infection. Because results of COVID-19 diagnostics were known during the ED visit and these patients differed from classical sepsis patients [33], sensitivity analyses were performed by excluding COVID-19 patients.

### 2.5. Statistical Analyses

Normally distributed data are presented as means with standard deviation (SD) and skewed data as median with interquartile ranges (IQR). Discrete values are shown as counts with percentages. For comparison of descriptive categorical variables, the chi-square test was used. In the case of continuous variables, the Student’s *t*-test or Mann–Whitney U test was performed for comparison of the 2 groups. For multiple group comparisons, Kruskal–Wallis test was used in combination with Dunn’s test for post hoc analyses. An optimal cut-off value for PSP was calculated by Youden’s index. The ability to discriminate between several groups was tested by constructing receiver operating characteristic (ROC) analyses with area under the curve (AUC). To investigate the additional value of PSP to current clinical practice, a multivariate binary logistic model was constructed with corresponding positive predictive value (PPV) and negative predictive value (NPV). This final model was constructed via backward stepwise selection. All statistical analyses were performed in SPSS version 26.0 for Windows (IBM Corp, Armonk, New York, NY, USA). All *p*-values < 0.05 were considered statistically significant.

## 3. Results

### 3.1. Baseline Characteristics

The final study population consisted of 156 patients, of which 56 patients (35.9%) had no infection, 74 (47.4%) had uncomplicated infection, and 26 (16.7%) patients had sepsis (Figure 1).

Table 1 shows the baseline characteristics of the cohort. Based on the qSOFA score, patients with sepsis were sicker than in the other two groups. Additionally, patients with sepsis were more frequently admitted to hospital (84.6% vs. 62.5% and 77.0%) and were more often immunocompromised (50.0% vs. 39.3% and 34.2%). The majority of patients presented at the general internal medicine department (34.6%). The most common source of infection was a viral systemic infection (including COVID-19) (*n* = 39, 25.0%), while urinary tract infections were the most frequent bacterial infection source (*n* = 16.7%).

### 3.2. Biomarker Distribution

Serum levels of WBC, CRP, and PSP are shown in Figure 2A–C. The median value for PSP (IQR) in the no infection, uncomplicated infection, and sepsis groups was 124.0 (79.0–205.0), 167.5 (94.5–310.3), and 182.0 (71.3–600.0) ng/mL, respectively; for CRP was 72.0 (20.3–128.0), 82.5 (31.5–146.3), and 80.0 (43.5–169.8), respectively; and for WBC was 8.5 (5.9–12.2), 9.4 (5.8–13.6), and 10.0 (5.8–15.6), respectively. There were no significant differences for any of these markers. Sensitivity analyses without COVID-19 patients are shown in Figure 2D–F. There were significant differences for PSP between the sepsis and no infection group (*p* = 0.035), but PSP was not able to discriminate between sepsis and uncomplicated infection (*p* = 0.43). There were no statistical differences for the three groups concerning WBC and CRP. Given the limitation of the abioSCOPE in measuring above 600 ng/mL, PSP was also analyzed dichotomously. The optimal cut-off value for PSP was 500 ng/mL. Significantly more patients had high PSP values in the sepsis group when compared to patients without infection (30.8% vs. 10.5%, *p* = 0.022) or uncomplicated infection (30.8% vs. 12.3%, *p* = 0.032). After exclusion of COVID-19, the optimal cut-off value was 199 ng/mL, and patients with sepsis had significantly higher values than patients with uncomplicated infection (60.0% vs. 35.0%, *p* = 0.049) and no infection (60.0% vs. 28.1%, *p* = 0.011).

### 3.3. Discrimination between No Sepsis and Sepsis

Next, the “no infection” group and “uncomplicated infection” group were merged into a “no sepsis” group to investigate the discriminative value of PSP for sepsis specifically. Figure 3A shows ROC curves for PSP, CRP, and WBC for the whole cohort.

All biomarkers showed little discriminative value and did not differ significantly from each other. After exclusion of COVID-19 patients, PSP scored a higher AUC when compared to CRP and WBC, although an AUC of 0.65 is still low. These ROC curves with corresponding AUC are shown in Figure 3B.

To investigate the additional value of PSP in current practice, a multivariate model was constructed. Potential predictors were chosen based on potential clinical relevance. In total, nine variables were included in the model (age, sex, immunocompromised status, Charlson Comorbidity Index, diabetes, COVID-19 infection, CRP, WBC, and PSP). Via backward selection, only three variables remained clinically relevant, including age, COVID-19 infection, and PSP. The AUC of this model was 0.69 with a negative predictive value of 84.4% and positive predictive value of 100% (Figure 4).

### 3.4. False-Positive and False-Negative Patients

To identify which patient characteristics may influence PSP performance, all false-positive and false-negative patients were analyzed. In Table 2, characteristics of the false-positive patients (high PSP values (≥199 ng/mL) but no convincing evidence for infection) are shown.

None of these patients was younger than 50 years old, and 10/15 (66.7%) were immunocompromised. For oncological patients, 4/5 (80%) appeared to suffer from auto-immune-mediated adverse effects from their immunotherapy, while the two hematological patients suffered from graft versus host disease (GvHD). CRP was <100 mg/mL for 9/15 (60.0%) patients, and leukocytes were within normal range (4.0–12.0 × 10^9^/L) for 8/15 (53.3%) patients.

Characteristics for false-negative patients (PSP < 199 ng/mL with sepsis) are described in Table 3. Most patients were relatively young, with 9/13 (69.2%) being younger than 50 years old. Of these patients, 6/13 (46.2%) were immunocompromised, and 5/13 (38.5%) suffered from COVID-19. Observing WBC and CRP values, 8/13 (61.5%) patients had WBC values within normal range, and 5/13 (38.5%) had CRP values <100 mg/mL. Strikingly, there were four patients with bacteremia, and, in all these patients, CRP was >100 mg/mL, while WBC values were abnormal in 3/4 (75%) of patients.

## 4. Discussion

This is the first study to investigate the prospective usability of point-of-care PSP measuring for diagnosing sepsis at the ED. Patients with sepsis had, on average, higher PSP values than patients without infection or patients with uncomplicated infection. When compared with other frequently used biomarkers, such as WBC and CRP, all markers performed poorly for the discrimination of sepsis versus non-sepsis patients. After exclusion of COVID-19 patients, PSP performed slightly better than CRP or WBC. In addition, in a multivariate model, PSP had significant additional value for current clinical practice to diagnose sepsis, but overall discriminative value was fairly low.

A couple of other studies have investigated the diagnostic value of PSP in the context of sepsis before. The study of Guadiana et al. is most comparable with the current study, since both studies investigated the diagnostic value of PSP for sepsis in an ED cohort of patients suspected of an infection [24]. However, Guadiana et al. showed excellent performance by PSP in diagnosing sepsis (ROC AUC = 0.872), in contrast with the current study. There are several explanations for these disparities. First of all, Guadiana et al. defined infection based on a majority rule among two physicians, and sepsis was based on sepsis-3 criteria (SOFA score). However, SOFA score is not useful in an ED setting, raising questions about the validity of this definition for sepsis in the ED. Additionally, we focused on all types of infection, while Gaudiana et al. limited their cohort to bacterial infection. Sepsis is not limited to bacterial infections, and, therefore, it is preferable to include all types of infection for diagnosing sepsis. Third, it is possible that the population of the UMC Utrecht consisted of more complicated patients than the population in the study of Guadiana et al. In the UMC Utrecht, approximately 40% of patients suspected of an infection were immunocompromised, and, due to the academic character of this hospital, patients are given highly complex care such as organ/stem cell transplantation, immunotherapy, and treatment for rare auto-immune diseases. Lastly, there is a remarkable difference between the median values for PSP in non-infectious patients (23 vs. 132 ng/mL), which might explain disparities in diagnostic performance. This discrepancy is most likely explained by differences in hospital population.

Another study investigated PSP in an unselected ICU cohort with impressive results [22]. PSP scored an AUC of 0.927 for diagnosing sepsis and was able to discriminate between sepsis and non-infective systemic inflammatory response (AUC 0.955). There are several explanations for these different outcomes. First of all, an ICU cohort is incomparable with an ED setting. Patients at the ICU are likely to have more inflammation and are sicker than at the ED [34,35]. Additionally, Llewelyn et al. performed the study on an unselected cohort, while the current study was performed for patients that were suspected of an infection. The latter would be more clinically useful, since patients without possible infections would not benefit from PSP point-of-care measuring. Lastly, also for this study, it is possible that the hospital populations were incomparable.

Our study has several strengths. First of all, PSP was measured prospectively within the structure of the SPACE cohort, resulting in robust data with well-defined outcomes. Additionally, PSP was measured by a point-of-care method, facilitating rapid potential clinical implementation. Additionally, MEWS was used to define sepsis, which is in line with the latest guidelines, enabling us to define sepsis at the ED based on standardized criteria [36]. Lastly, our study population was clinically more relevant than populations of studies that investigated unselected cohorts, since measuring biomarkers in patients that have no suspicion of infection is useless.

Our study has several limitations as well. Given the specific hospital population, our results might not be generalizable to hospitals with less complex patients. Second, this study was performed during the COVID-19 pandemic, resulting in a relatively large proportion of patients with a viral infection. Although COVID-19 might predispose patients to sepsis, there are some important differences between COVID-19 and sepsis [33]. For example, the hematological phenotype of COVID-19-induced coagulopathy differs significantly from sepsis-induced coagulopathy [37]. In addition, while cytokine storm plays a central role in sepsis, immunosuppression is the major problem in COVID-19 [38]. This statement is underlined by the relatively high percentage of COVID-19 patients in the false-negative group of our cohort. To resolve this issue, sensitivity analyses were performed without COVID-19 patients. Indeed, this improved the AUC of PSP but still did not result in a good performance. Third, Abionic’s abioSCOPE platform is only able to measure PSP up to 600 ng/mL [29]. As can be seen in our data, especially in the sepsis group, there were many patients with values >600 ng/mL. The exact PSP values for these patients are unknown, and, therefore, the statistical tests for continuous measurements, such as the Kruskal–Wallis test, were based on 600 ng/mL for these patients, while the actual values were higher. It could be that statistical differences were erased due to this limitation. To resolve this, we also used dichotomized values for PSP based on Youden’s index, the value of which was 199 ng/mL (after excluding COVID-19 patients), in line with previous literature [39]. In addition, PSP was compared with CRP and WBC but not with procalcitonin (PCT). We intended to compare PSP with routinely measured markers. In accordance with the latest Dutch sepsis guidelines, PCT is not routinely measured in Dutch hospitals, since it has no additional value when compared with CRP [36,40]. Therefore, this was not included in this study. A final limitation is that we used single PSP measurements. Some studies showed promising results using serial measurements of PSP and corresponding dynamic analyses [39]. This approach might be more usable, especially in complex hospital populations. Nevertheless, serial measurements are not suitable in an ED setting and can only be performed within hospital wards and/or ICU.

Concerning the interpretation of our results, it seems that point-of-care PSP measuring has additional value for current clinical practice, in contrast to CRP and WBC. However, overall discriminative value remained low. Biomarkers tend to perform well in unselected cohorts, but, from a clinical perspective, a good biomarker should be able to diagnose sepsis for patients with possible infection. Given the very high positive predictive value of the multivariate model, PSP might be used to rule in sepsis. Observing the characteristics of the false-positive and false-negative patients, older patients tend to have higher values than younger patients. Age was also included in our multivariate model, enforcing the positive correlation between PSP and age. This correlation was previously described by Zhu et al. [41], implying future studies should consider correcting for age when measuring PSP. Additionally, PSP is known to be increased in a variety of malignant diseases, which might hamper the use of PSP as sepsis marker in oncological patients [42,43]. Indeed, in our cohort, there was a lack of diagnostic value for oncological and hematological patients considering the overrepresentation of these patients in the false-positive group. A previous study confirms these findings in neutropenic patients specifically [26]. Most of the false-positive patients in the current study suffered from auto-immune-mediated inflammation. Previous literature showed PSP to be involved in the pathogenesis of auto-immune diseases, such as rheumatoid arthritis and inflammatory bowel disease, which affirms that PSP can be elevated in patients with auto-immune disorders as well [44,45].

Altogether, we conclude that PSP is probably limited in specific patient populations for diagnosing sepsis. Populations with high numbers of oncological patients and patients with auto-immune conditions seem to be too complex for PSP measuring in the context of sepsis. It is possible that PSP, above all, is mainly a marker of general inflammation instead of sepsis specifically. Therefore, future studies should focus on less complicated hospital populations with suspicion of infection. In such a cohort, it would be interesting to investigate the use of serial PSP measurements in combination with other clinical features to investigate its full potential in multivariate models.

## Figures and Tables

**Figure 1 pathogens-11-00559-f001:**
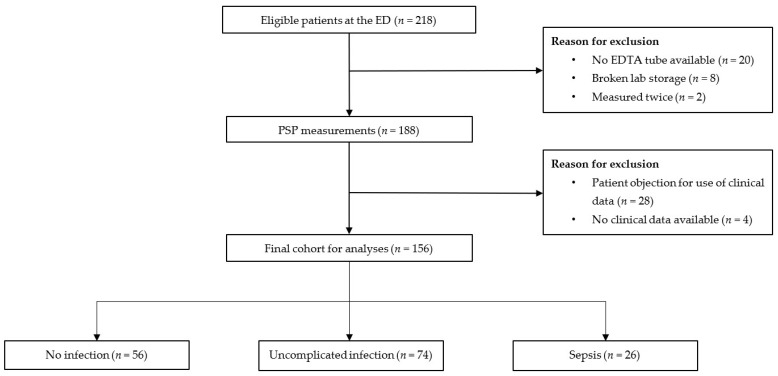
Flowchart of patient inclusion.

**Figure 2 pathogens-11-00559-f002:**
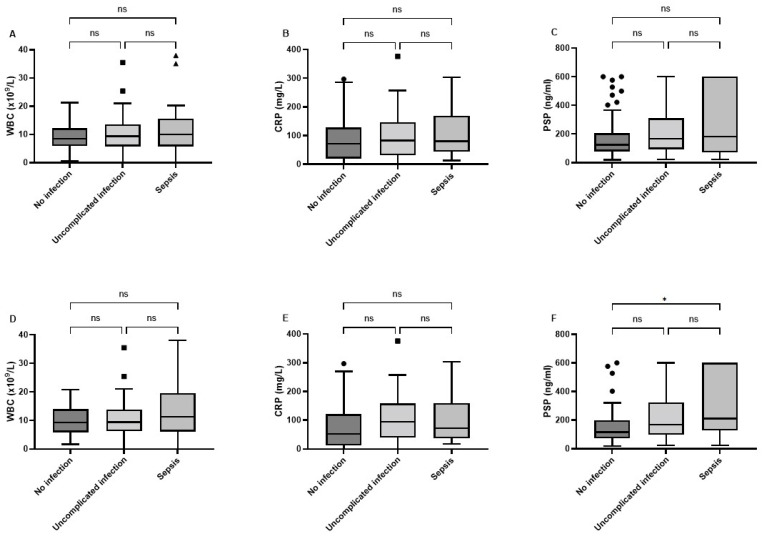
Boxplot distributions of WBC, CRP, and PSP in the whole cohort (**A**–**C**) after exclusion of COVID-19 patients (**D**–**F**). Statistically there were no differences between the groups for all three biomarkers in the whole cohort. There were significant differences between the three groups for PSP but not for WBC and CRP. The asterisk indicates significant difference (*p* < 0.05), whereas “ns” indicates not significant (*p* > 0.05).

**Figure 3 pathogens-11-00559-f003:**
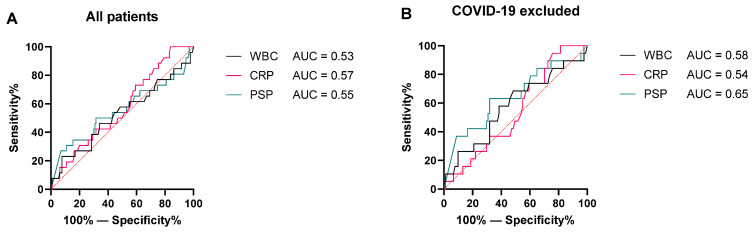
ROC analyses of WBC, CRP, and PSP. Over the total cohort (**A**), there was no clear difference between the three biomarkers. After excluding COVID-19 patients (**B**), PSP discriminated better between sepsis and non-sepsis patients when compared with WBC and CRP.

**Figure 4 pathogens-11-00559-f004:**
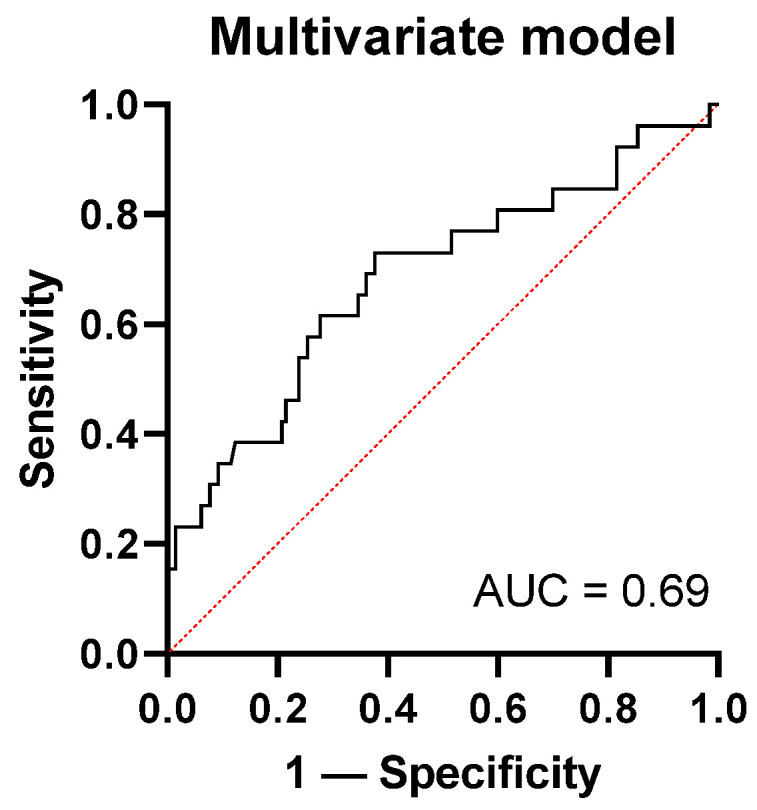
ROC analyses of our multivariate model, including the variables PSP, age, and COVID-19 infection. AUC of this model was 0.69 for discriminating between sepsis and no sepsis.

**Table 1 pathogens-11-00559-t001:** Baseline characteristics of cohort.

	Total (*n* = 156)	No Infection (*n* = 56)	Uncomplicated Infection (*n* = 74)	Sepsis (*n* = 26)
Demographics				
Age	60.0 (44.5–73.0)	55.0 (47.0–68.0)	66.0 (48.0–74.0)	54.0 (27.0–77.3)
Sex (M) (%)	82 (52.6)	28 (50.0)	42 (56.8)	12 (46.2)
Vital parameters				
Temperature (°C)	38.0 (37.0–38.9)	37.4 (36.9–38.4)	37.9 (36.8–38.4)	39.2 (38.9–39.6)
Heart rate (/min)	95.0 (82.0–110.0)	92.5 (80.0–107.5)	89.5 (80.0–105.0)	111.5 (104.5–117.8)
Respiratory rate (/min)	18.0 (15.8–22.0)	16.0 (14.0–19.0)	18.0 (15.0–22.0)	22.0 (19.5–30.5)
Systolic blood pressure (mmHg)	127.0 (112.0–144.0)	126.0 (113.5–142.0)	128.0 (112.0–148.0)	128.0 (105.0–153.3)
Diastolic blood pressure (mmHg)	71.0 (62.0–80.0)	74.0 (65.0–80.0)	69.0 (62.0–80.0)	69.0 (49.8–77.8)
Glasgow Coma Scale (EMV)	15.0 (15.0–15.0)	15.0 (15.0–15.0)	15.0 (15.0–15.0)	15.0 (14.0–15.0)
qSOFA score				
qSOFA = 0 (%)	102 (65.4)	41 (73.2)	52 (70.3)	9 (34.6)
qSOFA = 1 (%)	39 (25.0)	12 (21.4)	18 (24.3)	9 (34.6)
qSOFA = 2 (%)	14 (9.0)	3 (5.4)	3 (4.1)	8 (30.8)
qSOFA = 3 (%)	1 (0.6)	0	1 (1.4)	0
Hospitalization characteristics				
Admitted to hospital (%)	114 (73.1)	35 (62.5)	57 (77.0)	22 (84.6)
Length of stay (days)	4.0 (3.0–8.0)	4.0 (2.0–6.0)	5.0 (3.0–8.0)	4.0 (2.3–8.8)
Immunocompromised (%)	60 (38.5)	22 (39.3)	25 (34.2)	13 (50.0)
Charlson Comorbidity Index	4.0 (2.0–6.8)	3.0 (2.0–6.0)	5.0 (3.0–7.0)	3.0 (1.0–6.3)
Specialism				
General internal medicine (%)	54 (34.6)	15 (26.8)	28 (37.8)	11 (42.3)
Nephrology (%)	24 (15.4)	8 (14.3)	12 (16.2)	4 (15.4)
Hematology (%)	27 (17.3)	10 (17.9)	13 (17.6)	4 (15.4)
Oncology (%)	30 (19.2)	12 (21.4)	16 (21.6)	2 (7.7)
Rheumatology/Immunology (%)	8 (5.1)	5 (8.9)	1 (1.4)	2 (7.7)
Other (%)	13 (8.3)	6 (10.7)	4 (5.4)	3 (11.5)
Infection				
Lower respiratory tract (%)	18 (11.5)	-	9 (12.2)	4 (15.4)
Intra-abdominal (%)	20 (12.8)	-	12 (16.2)	1 (3.8)
Urinary (%)	26 (16.7)	-	16 (21.6)	7 (26.9)
Skin/soft tissue (%)	7 (4.5)	-	2 (2.7)	1 (3.8)
Viral systemic infection (%)	39 (25.0)	-	24 (32.4)	9 (34.6)
Cardiovascular (%)	4 (2.6)	-	2 (2.7)	2 (7.7)
Other (%)	16 (10.3)	-	9 (12.2)	2 (7.7)

**Table 2 pathogens-11-00559-t002:** Characteristics of false-positive patients.

Patient No.	Gender	Specialism	Age	Immunocompromised	Final Diagnosis	WBC (×10^9^/L)	CRP (mg/mL)	PSP (ng/mL)
1.	Male	Oncology	87	No	Auto-immune pneumonitis	14.10	85	202
2.	Male	Nephrology	73	Yes	Unknown	15.00	11	403
3.	Male	Oncology	67	No	Auto-immune gastroenteritis	14.10	31	472
4.	Male	Nephrology	57	Yes	Pericarditis	6.50	286	367
5.	Female	Hematology	61	Yes	Graft versus host disease	1.70	81	500
6.	Male	Hematology	53	Yes	Graft versus host disease	9.50	179	247
7.	Female	Nephrology	48	Yes	Unknown	4.10	7	242
8.	Male	Infectiology	50	No	Unknown	10.80	24	577
9.	Male	Nephrology	63	Yes	Unknown	13.60	49	320
10.	Female	Oncology	52	Yes	Unknown	5.90	159	287
11.	Female	Oncology	53	No	Auto-immune ileocolitis	20.10	114	528
12.	Female	Nephrology	58	Yes	Unknown	8.90	3	206
13.	Male	Oncology	81	No	Auto-immune pneumonitis	8.70	140	422
14.	Female	Nephrology	54	Yes	Unknown	18.40	140	601
15.	Female	Nephrology	59	Yes	Adverse effect eplerenon	21.30	0	601

**Table 3 pathogens-11-00559-t003:** Characteristics of false-negative patients.

Patient No.	Gender	Specialism	Age	Immunocompromised	Final Diagnosis	Microbiological Culture	WBC (×10^9^/L)	CRP (mg/mL)	PSP (ng/mL)
1.	Female	Infectiology	47	No	Viral systemic	Coronavirus (Throatswab)	7.10	233	48
2.	Female	Hematology	20	Yes	Viral systemic	Coronavirus (Throatswab)	6.70	136	63
3.	Female	Immunology	21	Yes	Lower respiratory tract	-	8.60	71	42
4.	Female	Hematology	27	Yes	Lower respiratory tract	-	1.50	172	74
5.	Male	General internal medicine	24	No	Viral systemic infection	Coronavirus (Throatswab)	3.10	13	44
6.	Male	General internal medicine	22	No	Other	*Streptococcus pneumonia* (Blood)	35.10	169	92
7.	Female	General internal medicine	29	Yes	Viral systemic	Coronavirus (Throatswab)	4.20	46	26
8.	Female	Oncology	41	No	Urinary tract	*Escherichia coli* (Urine)	6.10	31	134
9.	Male	Hematology	68	Yes	Urinary tract	*Escherichia coli* (Blood)	0.00	189	111
10.	Female	General internal medicine	78	No	Cardiovascular	*Staphylococcus aureus* (Blood)	9.90	159	23
11.	Female	Infectiology	35	Yes	Viral systemic	*Cytomegalovirus* (Blood)	6.00	126	141
12.	Male	General internal medicine	84	No	Viral systemic	Coronavirus (Throatswab)	9.40	99	164
13.	Male	General internal medicine	59	No	Cardiovascular	*Staphylococcus aureus* (Blood)	19.70	224	126

## Data Availability

The data presented in this study are available on reasonable request from the corresponding author. The data are not publicly available due to privacy considerations.

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
