# Peer review of "Pancreatic Stone Protein as a Biomarker for Sepsis at the Emergency Department of a Large Tertiary Hospital"

_pathogens, 2022, doi:10.3390/pathogens11050559_

Round 1

Reviewer 1 Report

The study idea is nice and the methodology is well performed. The results are well described and the conclusions are in line with the study idea. I have only a comment: what is the role of procalcitonin? Please improve this aspects in the methodology and discussion. 

Author Response

Response to reviewer 1

The study idea is nice and the methodology is well performed. The results are well described and the conclusions are in line with the study idea. I have only a comment: what is the role of procalcitonin? Please improve this aspects in the methodology and discussion.

We thank the reviewer for his/her critical appraisal. Our intention was to compare PSP with routinely measured biomarkers for sepsis. Since procalcitonin (PCT) is not routinely measured in Dutch hospitals, and the UMC Utrecht specifically, this was not included in our analyses. Overall, clinical performance of PCT is comparable to CRP in the context of sepsis. Therefore, it is our believe that measuring procalcitonin is costly and does not add much to current clinical practice. Although we agree that it would have been interesting to compare PSP with PCT if PCT would have been easily available, this was not the case. To be clear on this subject we added the following parts to our methods and discussion section:

Line 109-110: “PSP values were compared with routinely measured biomarkers CRP and WBC.“

Line 285-289 : “Also, PSP was compared with CRP and WBC, but not with procalcitonin (PCT). We intended to compare PSP with routinely measured markers. In accordance with the latest Dutch sepsis guidelines, PCT is not routinely measured in Dutch hospitals, since it has no additional value when compared with CRP [36,40]. Therefore, this was not included in this study.”

Reviewer 2 Report

Dear Authors, Please improve and extent your introduction Please add more conclusion to your article Kind regards

Author Response

Response to reviewer 2

Please improve and extent your introduction Please add more conclusion to your article.

Thank you for these recommendations. We propose the following extension for the introduction. If the reviewer would like other sections of the introduction to be extended, we would like to hear which sections should be adjusted.

The conclusion was elaborated as well.

Introduction (line 59-71)

More importantly in the context of sepsis, there are several findings that indicate that PSP is involved in the defense mechanisms during the early phase of sepsis. For instance, PSP was elevated in patients with posttraumatic infections and infectious complications of burn wounds [18,20]. In these studies, PSP showed to be related to the severity of inflammation and was able to activate neutrophil granulocytes by upregulating activation markers CD11b and CD62L [18]. Besides neutrophil activation, PSP by itself is known to have antibacterial functions as well. PSP is able to induce bacterial aggregation which might be beneficial to prevent bacteria penetrating the intestinal barrier in the gut [21]. At last, PSP levels have shown to start increasing up to 72 hours before clinical symptoms of sepsis appeared. Therefore, PSP might be helpful to diagnose sepsis before patients start to get critically ill [20]. Altogether, the association with severity of inflammation, antimicrobial involvement, and the capabilities to rise during early infection, make PSP an interesting biomarker for early sepsis recognition.

Conclusion (line 315-320)

Populations with high rates of oncological patients, and patients with auto-immune conditions seem to be too complex for PSP measuring in the context of sepsis. It is possible that PSP, above all, is mainly a marker of general inflammation instead of sepsis specifically. Therefore, future studies should focus on less complicated hospital populations with suspicion of infection. In such cohort it would be interesting to investigate the use of serial PSP measurements…..